# Improving Phylogenetic Signals of Mitochondrial Genes Using a New Method of Codon Degeneration

**DOI:** 10.3390/life10090171

**Published:** 2020-08-30

**Authors:** Xuhua Xia

**Affiliations:** 1Department of Biology, University of Ottawa, 30 Marie Curie, Ottawa, ON K1N 6N5, Canada; xxia@uottawa.ca; 2Ottawa Institute of Systems Biology, 451 Smyth Road, Ottawa, ON K1H 8M5, Canada

**Keywords:** codon degeneration, phylogenetic conflict, mtDNA, deep phylogeny, ratite

## Abstract

Recovering deep phylogeny is challenging with animal mitochondrial genes because of their rapid evolution. Codon degeneration decreases the phylogenetic noise and bias by aiming to achieve two objectives: (1) alleviate the bias associated with nucleotide composition, which may lead to homoplasy and long-branch attraction, and (2) reduce differences in the phylogenetic results between nucleotide-based and amino acid (AA)-based analyses. The discrepancy between nucleotide-based analysis and AA-based analysis is partially caused by some synonymous codons that differ more from each other at the nucleotide level than from some nonsynonymous codons, e.g., Leu codon TTR in the standard genetic code is more similar to Phe codon TTY than to synonymous CTN codons. Thus, nucleotide similarity conflicts with AA similarity. There are many such examples involving other codon families in various mitochondrial genetic codes. Proper codon degeneration will make synonymous codons more similar to each other at the nucleotide level than they are to nonsynonymous codons. Here, I illustrate a “principled” codon degeneration method that achieves these objectives. The method was applied to resolving the mammalian basal lineage and phylogenetic position of rheas among ratites. The codon degeneration method was implemented in the user-friendly and freely available DAMBE software for all known genetic codes (genetic codes 1 to 33).

## 1. Introduction

In a multiple sequence alignment, there are historical signals, such as the number of nucleotide substitutions, that are typically proportional to the divergence time, and non-historical signals that are typically not proportional to the divergence time [1]. Non-historical signals include compositional bias [2,3,4] and conflicting signals between codons and amino acids (AAs) [2,3,5,6]. Codon degeneration, when properly implemented, can minimize or eliminate these non-historical signals in aligned sequences [7]. I will outline these two sources of undesirable signals, detail a codon degeneration method, and apply the method to mammalian and avian mitochondrial sequences to resolve (1) the phylogeny of basal eutherian lineages and (2) the phylogenetic position of rheas among ratites.

### 1.1. Nucleotide Composition Bias

Nucleotide composition bias refers to the phenomenon in which distantly related taxa share similar nucleotide frequencies, leading to a spurious similarity between such taxa [8,9]. The problem is particularly serious when one aims to construct a universal tree [10].

Nucleotide composition bias can arise via either shared selection or shared mutation. For example, to stabilize the stem-loop secondary structure in their rRNAs, thermophilic bacteria tend to have not only GC-rich stems but also longer stems, regardless of their phylogenetic affinity [11]. Such convergent evolution due to shared high ambient temperature has long been identified as a potential source of homoplasy [12]. In particular, mesophiles, such as *Deinococcus* and *Bacillus* species, are relatively AU-rich in their 16S rRNA, in contrast to thermophiles, such as *Aquifex, Thermotoga,* and *Thermus* species (Figure 1A). Foster [13] used this example to illustrate the importance of accommodating such composition heterogeneity in phylogenetics. Conventional phylogenetic methods that do not accommodate such compositional heterogeneity consistently group the two mesophiles together (Figure 1B). However, when additional parameters are allowed to model lineage-specific nucleotide frequencies, a correct topology (Figure 1C) was recovered. For this reason, much effort has been spent in the search for efficient methods to model a nonstationary substitution process [13,14,15,16,17,18].

Nucleotide and codon degeneration [7,20,21] offers a simple alternative to modeling a nonstationary substitution process. After degenerating the sequences into purine and pyrimidine, the correct tree was recovered by using either likelihood methods or a distance-based method using GTR or simpler substitution models (Figure 1C). Note that using LogDet [8] and paralinear [9] distances, as implemented in DAMBE [22], invariably led to the wrong tree in Figure 1B instead of that in Figure 1C when the sequences were not degenerated. Thus, LogDet and paralinear distances do not accommodate compositional heterogeneity as claimed.

Another selection-mediated source of composition bias is tRNA-mediated selection on codon usage bias. Bacteriophages in *Escherichia coli* exhibit similar codon usage to its host genes (especially highly expressed ones), regardless of their phylogenetic affinity, presumably to take advantage of differential tRNA availability in the host tRNA pool [23,24,25]. Such tRNA-mediated composition bias also occurs in mitochondrial sequences. For example, some bivalve and chordate species have two Met tRNAs, tRNA^Met/CAU^ and tRNA^Met/UAU^, where CAU and UAU are anticodons, to translate Met AUG and AUA codons. In contrast, most other species only have a single tRNA^Met/CAU^ to translate both Met codons AUG and AUA, where the nucleotide C in the first anticodon site is modified to pair with both A and G [26]. The independent gain of tRNA^Met/UAU^ has resulted in the convergent increase of AUA codon usage in bivalve and cordate species [27,28].

In addition to responding to the shared selection, composition bias can also arise through shared mutation bias [29]. Diverse parasitic bacterial lineages are almost invariably AT-rich [30,31], presumably because spontaneous mutations tend to be AT-biased [32,33]. This also occurs in ancient DNA with differential nucleotide decay [34]. Mitochondrial [35] and bacterial nuclear genomes [36] both exhibit strand bias, where a gene that has switched strand during evolution experiences dramatically different mutation spectra and accumulates substitutions rapidly, leading to an extraordinarily long branch involving the strand-switched gene [27]. In particular, composition bias often changes direction rapidly [37,38], and is therefore not proportional to time.

### 1.2. Conflicting Signal between Nucleotide and Amino Acid Sequences

There are often phylogenetic conflicts between nucleotide-based and AA-based analyses [2,3,5]. Several codon families contribute to this discrepancy, and I will illustrate this with five examples. First, nearly all genetic codes (except for genetic codes 3 and 23) have TTR and CTN encoding amino acid Leu, and TTY encoding amino acid Phe (Figure 2A), where R stands for purine, N for any nucleotide, and Y for pyrimidine. Leu codons TTA and TTG are more similar to Phe codons TTC and TTT than to synonymous Leu codons CTC and CTT. If we use the match/mismatch score matrix in Figure 2F, the alignment score between nonsynonymous Leu codon TTR and Phe codon TTY is 15 but the alignment score between the synonymous TTR and CTY is only 0 (Figure 2A). The alignment score is an index of sequence similarity. Two aligned sequences with a large alignment score are more similar to each other than two with a small alignment score. Two sequences with a high alignment score are also expected to have a smaller evolutionary distance between them than two sequences with a low alignment score. Thus, at the AA level, sequences L1 to L6 (Figure 2A) are identical but differ from sequences F1 and F2. However, at the nucleotide level, L5 and L6 are more similar to F1 and F2 than to L2 and L4 (Figure 2A).

Second, in most genetic codes (except for genetic codes 2, 5, 9, 13, 14, 21, 24, and 33), AGR and CGN encode Arg, and AGY encodes Ser. Arg codon AGR is more similar to Ser codons AGY (with an alignment score of 30; Figure 2B) than to synonymous Arg codons CGC and CGT (with an alignment score of −30; Figure 2B). Again, at the AA level, sequences R1 to R6 are identical to each other but differ from S1 and S2 (Figure 2B). In contrast, at the nucleotide level, sequences R5 and R6 are more similar to S1 and S2 than they are to R2 and R4 (Figure 2B), leading to conflicts between the AA signals and nucleotide signals.

Examples 3 to 5 are from specialized genetic codes. In genetic code 25 (Candidate Division SR1 and Gracilibacteria Code), TGA is a Gly codon instead of a stop codon, as in the standard code. This Gly codon TGA is more similar to Trp codon TGG (with an alignment score of 60; Figure 2C) than to other synonymous Gly codons, with an alignment score varying from −30 to 30 (Figure 2C). In genetic codes 24 (Rhabdopleuridae Mitochondrial Code) and 33 (Cephalodiscidae Mitochondrial UAA-Tyr Code), AGG is a Lys codon, which is more similar to the Ser codon AGA (alignment score = 60) than to the synonymous codon AAA (alignment score = 30; Figure 2D). Finally, in genetic code 13, Gly codons AGA and AGG are more similar to Ser codon AGC and AGT, with an alignment score of 30, than to synonymous Gly codons GGC and GGT (with an alignment score of 0; Figure 2E). We need to find a codon degeneration method that will ideally achieve the objective of finding synonymous codons in Figure 2 that are more similar to each other than they are to nonsynonymous codons. If we designate a minimum similarity between synonymous codons as $S_{min.S}$ (which equals 0 for Leu codons in Figure 2A) and a maximum similarity between nonsynonymous codons as $S_{max.NS}$ (which equals 60 between Leu codons and Phe codons in Figure 2A), we wish to have a codon degeneration method that ideally yields $S_{min.S}>S_{max.NS}$.

The codon degeneration method I present here was previously implemented by myself for the standard genetic code [7]. I extended the implementation to support all known genetic codes summarized in Xia [39], plus two more recent codes, including 14 genetic codes that are specific for mitochondrial genomes. I applied the codon degeneration method to the analysis of mitochondrial sequences to (1) resolve basal eutherian lineages and (2) elucidate phylogenetic placement of rheas among ratites.

## 2. Materials and Methods

### 2.1. The “Principled” Codon Degeneration and Two Alternatives

One could perform three different types of codon degeneration to alleviate the problems caused by composition bias and conflict signals between nucleotide and AA sequences. First, we may just degenerate the third codon position (Figure 3), which has been used to alleviate the phylogenetic bias caused by divergent sequences with similar nucleotide compositions [40,41,42]. For example, the sequences from arthropod taxa [20] differ significantly in the nucleotide frequencies, with GC content at the third codon position (GC_3_%) varying from 37.88% to 80.42% in the three ostracods and from 24.10% to 64.40% in arachnids [7]. The degeneration also reduced conflicting signals between nucleotide similarity and AA similarity. Take the Leu and Phe codons Figure 3A for example. $S_{min.S}=22$ and $S_{max.NS}=30$. Although this falls short of achieving $S_{min.S}>S_{max.NS}$, it is much better than the nondegenerated case where $S_{min.S}=0$ and $S_{max.NS}=60$ (Figure 2A). The $S_{min.S}$ and $S_{max.NS}$ values for all five illustrated cases in Figure 2 and Figure 3 are listed in columns 3 and 4 in Table 1. The difference between $S_{min.S}$ and $S_{max.NS}$ have all changed in the right direction, i.e., $S_{min.S}$ has increased and $S_{max.NS}$ has mostly decreased (Table 1).

The second type of codon degeneration, which we previously named “principled” [7], degenerates the two Leu codons TTA and TTG to YTR and Leu codons CTA, CTG, CTC, and CTT to CTN (Figure 4A). The conceptual principle is that any codon degeneration should not lead to a degenerated codon losing its AA identity. In other words, a degenerated codon should never include nonsynonymous codons. The operational principle of this codon degeneration is that for two synonymous codon subfamilies of different sizes (e.g., one with four codons and the other with two, as in the case of Leu codons in Figure 2A), codon positions 1 or 2 are degenerated only in the smaller codon subfamily. For example, CTN includes four codons and TTR includes two codons (i.e., the smaller of the two); therefore, the first codon position of only the smaller TTR family is degenerated to YTR. Note that degenerating the first codon position of CTN further to YTN, as in Regier et al. [20], would violate the conceptual principle because YTN encompasses both Leu and Phe codons. If two codon subfamilies differ in both the first and second codon positions, then they are treated as separate codon families and degenerated independent of each other. For example, Ser codons in the standard code is degenerated into AGY and UCN because any further degeneration would violate the conceptual principle.

The Leu codons YTR and CTN, as well as the Phe codon family TTY, were degenerated according to this “principled” codon degeneration (Figure 4A). Similarly, the two Arg subfamilies (Figure 2B) were degenerated to CGN and MGR (Figure 4B). Note that only the smaller subfamily has the first codon position degenerated to M (standing for either A or C). The two Gly subfamilies (Figure 4C) were degenerated to GGN and KGA (where K stands for either G or T). Again, only the smaller codon family has its first codon position degenerated. The two Lys subfamilies, one with two codons and one with a single codon AGG (Figure 2D), were degenerated to AAR and ARG (Figure 4D), with only the smaller codon subfamily (i.e., AGG) having its second codon position degenerated. Degenerating the larger subfamily AAR further to ARR would have violated the conceptual principle because ARR encompasses not only the three Lys codons but also the Ser codon AGA (Figure 4D). The two Gly subfamilies (Figure 4E) were degenerated to GGN and RGR. In short, the conceptual principle is maintained by sticking to the operational principle of degenerating the first or second codon position of the smaller codon subfamily. This codon degeneration function can be accessed in DAMBE by clicking “Sequences|Sequence manipulation|Degenerate synonymous codons”.

The benefit of the “principled’’ codon degeneration is clearly visible in $S_{min.S}$ and $S_{max.NS}$ (last column in Table 1) or by contrasting Figure 3 and Figure 4. The nucleotide similarities among synonymous codons have consistently increased and similarities between nonsynonymous codons have consistently increased. For example, before the codon degeneration, Leu codons TTA and TTG had a nucleotide similarity of 0 to synonymous Leu codons CTC and CTT, which is much smaller than that to the two nonsynonymous Phe codons TTC and TTT ($S_{max.NS}=60$; Table 1). After the “principled” codon degeneration, $S_{min.S}$ increased to 37 and $S_{max.NS}$ decreased to 22. Thus, the nucleotide similarity and AA similarity are no longer conflicting.

The third type of codon degeneration is used in Regier et al. [20] and violates the conceptual principles above. For example, it degenerates all six Leu codons in Figure 2A to YTN. This obscures the differences between the six Leu codons and the two Phe codons (TTY) because YTN encompasses both. With the “principled” codon degeneration, synonymous Leu codons are all more similar to each other than they are to the two nonsynonymous Phe codons (Figure 4A), with $S_{min.S}=37$ and $S_{max.NS}=22$ (Table 1). This third type of codon degeneration results in all six Leu codons and the two Phe codons having the same nucleotide similarity with $S_{min.S}=S_{max.NS}=37$. The same is true for the Arg codon family in Figure 2B. Regier et al. [20] degenerated all six Arg codons to MGN, which again violates the conceptual principle because MGN encompasses both the six Arg codons and the two Ser codons in Figure 2B. With the “principled” codon degeneration, the six synonymous Arg codons are more similar to each other than to the two Ser codons (Figure 4B), with $S_{min.S}=22$ and $S_{max.NS}=0$. The third type of codon degeneration renders $S_{min.S}=S_{max.NS}=22$. Furthermore, note that the codon sequences can no longer be translated back into amino acid sequences after this third type of codon degeneration. Losing codon identity represents a significant loss of evolutionary information since we can no longer perform codon-based analysis. As pointed out before [7], Miyata’s distance is 2.73 between Arg and Ser and 0.63 between Phe and Leu [43], and empirical data suggests that replacement with synonymous codons is more likely than between Arg and Ser or between Phe and Leu codons, according to Figure 13.1 in Xia [44]. Therefore, it is not a good idea to treat nonsynonymous codons as equivalent to synonymous codons.

### 2.2. Mitochondrial Data and Phylogenetic Analysis

I used two sets of mitochondrial sequences to evaluate the phylogenetic performance of the “principled” codon degeneration, addressing two phylogenetic problems. The first involved the resolution of basal eutherian lineages. I downloaded mitochondrial genomes from 11 mammalian species representing the basal eutherian lineages: *Sus scrofa* (mtDNA accession NC_000845), *Loxodonta africana* (NC_000934), *Equus caballus* (NC_001640), *Dasypus novemcinctus* (NC_001821), *Oryctolagus cuniculus* (NC_001913), *Artibeus jamaicensis* (NC_002009), *Orycteropus afer* (NC_002078), *Galeopterus variegatus* (NC_004031), *Mus musculus* (NC_005089), *Delphinus capensis* (NC_012061), and *Homo sapiens* (NC_012920). The 13 protein-coding genes were extracted using DAMBE [45]. Codon sequences were aligned against aligned AA sequences as an automated process in DAMBE using MAFFT [46,47] with the most accurate but slower LINSI option (“–localpair” and “–maxiterate = 1000”). Individually aligned sequences were then concatenated into the Appendix A. Individuals and their aligned lengths in the same order as in concatenated supermatrix were ATP6_ATP8 (882), COX1 (1551), COX2 (687), COX3 (783), CYTB (1143), ND1 (954), ND2 (1047), ND3 (351), ND4 (1377), ND4L (294), ND5 (1833), and ND6 (537).

The second phylogenetic problem was about the phylogenetic position of rhea in ratites. I used concatenated mitochondrial coding sequences from 11 mitochondrial genomes [48], including seven paleognathes: *Struthio camelus* (ostrich, GenBank ACCN: NC 002785), *Dromaius novaehollandiae* (emu, NC 002784), *Casuarius casuarius* (cassowary, NC 002778), *Apteryx haastii* (kiwi, NC 002782), *Dinornis giganteus* (extinct moa, NC 002672), *Rhea pennata* (rhea, NC 002783), *Eudromia elegans* (tinamou, NC 002772), and four neognathes: *Gallus gallus* (chicken, NC 001323), *Branta canadensis* (Canada goose, NC 007011), *Phoenicopterus roseus* (flamingo, NC 010089), and *Rhynochetos jubatus* (kagu, NC 010091). Coding sequences were extracted using DAMBE, individually aligned, and then concatenated. The supermatrix is included as the Appendix A

PhyML [19] was used for phylogenetic reconstruction using the GTR+Γ substitution model (the best model based on likelihood ratio tests or information-theoretic indices). The tree improvement option “-s” was set to “BEST” (best of NNI and SPR searches). The “-o” option was set to “tlr”, which optimized the topology, the branch lengths, and the rate parameters.

## 3. Results

### 3.1. Codon Degeneration Increased the Phylogenetic Resolution Power in Early Mammalian Lineages

The aligned mitochondrial sequences representing basal eutherian lineages were analyzed without (Figure 5A) and with the “principled” codon degeneration (Figure 5B). The two resulting topologies were identical. The topology was well corroborated using diverse data, and in particular, validated by the sharing of retroelements [49]. The two trees (Figure 5) differ in support values for internal nodes. The tree generated using the “principled” codon degeneration (Figure 5B) had substantially higher support values for some nodes than the tree generated without codon degeneration (Figure 5A).

### 3.2. Codon Degeneration Challenged the Conventional Phylogenetic Placement of Rhea

The phylogenetic position of flightless ratites and tinamous have recently been elucidated by using ancient DNA from museum specimens [50,51,52]. However, the phylogenetic position of rheas is inconsistent, being placed at the root or close to the root of ratites and tinamous in an analysis of mitochondrial genes [50,51], but closer to the emu–cassowary–kiwi clade in an analysis when nuclear genes are used [52]. Here, I show that the placement of rhea close to the root of the ratites and tinamous was due to composition bias that could be corrected using codon degeneration.

The phylogenetic analysis was again performed without (Figure 6A) and with the “principled” codon degeneration (Figure 6B). One may get an intuitive sense of the compositional bias by examining the proportion of nucleotide C (P_C_), which is the most abundant nucleotide in these avian mitochondrial genes in the sequences. P_C_, shown after each species name in Figure 6A, is higher in the four neognathes than the P_C_ in most paleognathes. *Rhea pennata* happens to have the highest P_C_, which spuriously increases its sequence similarity to the four neognathes and pull it toward the root. Furthermore, the four neognathes encode Leu mostly using CUN instead of UUR, with P_CUN_ = 0.8779, which is higher than that for the species (excluding *Rhea pennata*) in paleognathes (P_CUN_ = 0.8381). *Rhea pennata*, because of its C-richness, has P_CUN_ = 0.8841. This increases the chance of Leu at a site being encoded by CUN in *Rhea pennata* + neognathes, but by UUR in other paleognathes, further increasing the spurious sequence similarity between *Rhea pennata* and the four neognathes.

The phylogeny from the codon-degenerated sequences (Figure 6B) has *Rhea pennata* clustered together with emu (*Dromaius novaehollandiae*), cassowary (*Casuarius casuarius*), and kiwi (*Apteryx haastii*). The phylogeny based on AA sequences translated from the coding sequences also had these four species forming a monophyletic cluster. Furthermore, the phylogenetic relationship from nucleotide sequences without compositional bias also suggested a closer relationship between rhea and the kiwi+cassowary+emu clade [52,53]. These multiple lines of evidence suggest that the placement of rhea close to the root of paleognathes [50,51] was due to compositional bias that could be corrected by codon degeneration. Note that the phylogenetic placement of *Rhea pennata*
Figure 6B differs from recent publications [52,53] that have the phylogenetic positions of rhea and kiwi swapped. However, these two studies, albeit with an extensive data compilation and comprehensive data analysis, did not pay particular attention to composition bias, and simply asserted that the noncoding sequences they used were less subject to composition bias than coding sequences. Even if the assertion is true, it does not mean that noncoding sequences are immune to composition bias. There is strand-specific nucleotide bias in both nuclear and mitochondrial genomes [27,35,54] such that an inversion event leading to a sequence switching strands typically results in very different substitution patterns.

The phylogeny in Figure 6B is consistent with continental vicariance, as illustrated in Figure 7. The geophylogeny (mapping of a phylogeny onto geographic locations) was drawn using PGT software [55]. In the late Cretaceous period (Figure 7, inset A), Africa was separated from South American + Antarctic + Australasia, isolating the ostrich from the rest of the paleognaths (Figure 7). The small and nocturnal ancestor of kiwi should have diverged from the ancestor of the large and diurnal cassowary+emu+rhea clade. The subsequent separation of South American from Antarctica + Australasia resulted in (1) isolation of the rhea lineage from the cassowary + emu lineage and (2) isolation of the tinamou lineage from the moa lineage (Figure 7).

## 4. Discussion

### 4.1. When to Use Codon Degeneration?

The codon degeneration method can alleviate compositional bias and reduce differences in phylogenetic analysis from nucleotide and AA sequences when used properly in two specific scenarios. The first scenario is when the composition bias is such that remotely related taxa share similar nucleotide frequencies than closely related species. For example, a GC-rich gene may encode amino acid Leu using CUG, but this codon may change into UUG in a closely related but AT-rich gene. Biased mutation can change directions quite rapidly [11,37,56,57]. Degenerating the third position would remove the difference caused by mutation bias between two closely related species. The second scenario is when the same AA site in a set of aligned sequences is encoded by two blocks of codons, as in the case of Leu codons and Arg codons in the standard genetic code. As is illustrated in Figure 4 and Table 1, the principled codon degeneration reduces the difference between synonymous codons such that the difference in phylogenetic results between the nucleotide-based and AA-based analysis is reduced.

Codon degeneration would not be appropriate when reconstructing the phylogeny of closely related species. For closely related species, sister taxa typically share similar nucleotide frequencies, which are consequently also phylogenetically informative. Furthermore, if Leu at each site is encoded by either UUR or CUN, but never both (and if Arg at each site is encoded by either AGR or CGN, but never both), then the benefit of codon degeneration would be minimal and may not offset the cost of lost information. Typically, only highly diverged sequences may benefit from codon degeneration.

### 4.2. “Principled” Degeneration versus Degenerating the Third Codon Site Only

The “principled” codon degeneration aims to achieve two objectives: (1) minimize the composition bias and (2) remove conflicting signals between the nucleotide and AA sequences. Degenerating the third codon position should achieve the first objective; therefore, it is interesting to compare the two degeneration methods. I have added phylogenetic results (Figure 8) from sequences degenerated at the third codon only (which should remove most of the composition heterogeneity but does not remove the conflicting signals between the nucleotide and AA sequences). The phylogeny in Figure 8A (with degeneration at the third codon sites only) was comparable to that in Figure 5B (with “principled” codon degeneration). The tree topologies were the same, and the only major difference was the support value of 71 (red in Figure 8A) versus the corresponding value of 92 in Figure 5B.

The phylogeny in Figure 8B (with degeneration at the third codon sites only) was comparable to that in Figure 6B (with the “principled” codon degeneration). The topologies were again the same, with the only notable support value of 46 (red in Figure 8B) being substantially lower than the corresponding value of 74 in Figure 6B. The comparisons suggest that the “principled” codon degeneration was more preferable over degenerating the third codon sites only, although more empirical substantiation is needed.

### 4.3. The Serine Codon Family

The codon degeneration method cannot help with synonymous codons encoded by disjoint blocks of codons, such as Ser codons. Ser is encoded by TCN and AGY for most genetic codes. Sites with Ser codons may distort the phylogenetic signals at the nucleotide level if two closely related taxa happen to have TCN and AGY, respectively, at the same homologous codon site. No codon degeneration method makes two synonymous codons, such as TCN and AGY, more similar to each other than between two nonsynonymous codons, such as Ser codon AGY and Arg codon AGR.

The presence of TCN and AGY codons at the same codon site may cause conflict between nucleotide-based and AA-based analyses [2,3,5,6]. One way to avoid this problem is simply to remove such codon sites. DAMBE offers the option of simply removing sites containing both TCN and AGY codons before nucleotide-based analysis. The function can be accessed by clicking on “Sequence|Sequence manipulation|Remove sites with both UCN and AGY serine codons”. The function also provides an option to keep only those codon sites containing UCN and AGY such that it can be checked whether they contribute significant phylogenetic signals. The concatenated mammalian coding sequences (Appendix A) contain 40 codon sites with both UCN and AGY serine codons. I constructed a tree from these 40 codons. The tree shared only a single bipartition with the tree in Figure 5 out of eight bipartitions. This was similar to randomly generated sequences with the same nucleotide frequencies, indicating little information in those codon sites containing both UCN and AGY codons. However, removing these codon sites did not consistently increase the support values for the internal nodes (Figure 9). The phylogeny in Figure 9A was comparable to that in Figure 5A, with the former from sequences after removing the 40 codon sites featuring both UCN and AGY Ser codons, and the latter without removing them. No codon degeneration was done in both cases. The phylogeny in Figure 9B was comparable to that in Figure 5B, both with the “principled” degeneration but they differed in that the former removed the 40 codon sites and the latter did not. The topologies were all the same, and there was no consistent improvement in the support values in both comparisons.

The avian mitochondrial sequences (Appendix A) contained only 14 codon sites featuring both UCN and AGY codons. Removing them did not improve the phylogenetic resolution. Furthermore, there was variation in the Ser encoding in genetic codes 5, 9, 12, 14, 21, 24, and 33, i.e., the Ser codons were not limited to TCN and AGY, such that caution should be exercised when removing codon sites with both UCN and AGY codons because they may not be Ser codons.

## 5. Conclusions

Codon degeneration methods can improve the phylogenetic signals of highly divergent sequences. It should help to solve the difficult problem of resolving deep phylogeny.

## Figures and Tables

**Figure 1 life-10-00171-f001:**
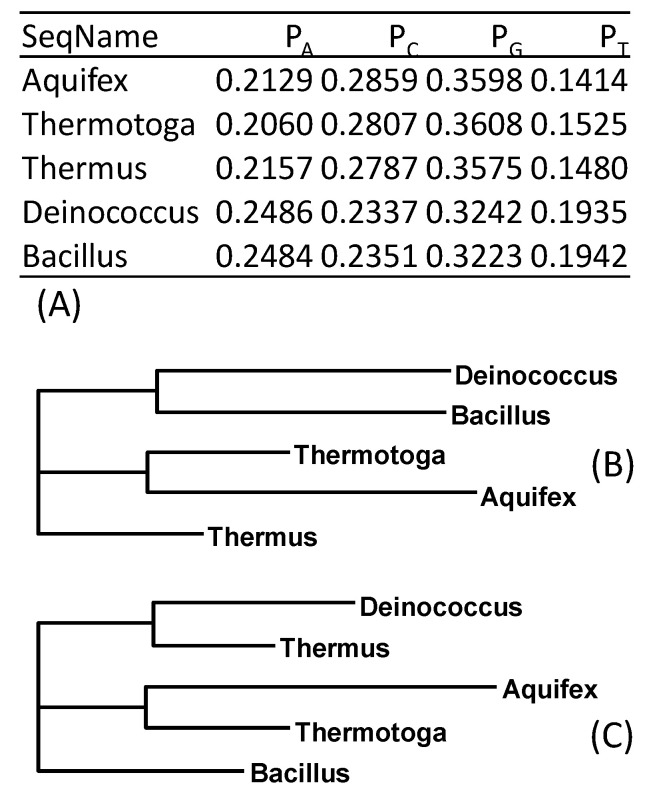
Similarity in nucleotide frequencies between *Deinococcus* and *Bacillus* favors them being clustered together. (**A**) Nucleotide frequencies of 16S rRNA from five prokaryotes [13]. (**B**) The same phylogenetic tree produced from software PhyML [19] with GTR with or without using a gamma distribution to accommodate the rate heterogeneity. (**C**) After degenerating the sequences to purines and pyrimidines, the correct phylogenetic tree was recovered from PhyML with the same options.

**Figure 2 life-10-00171-f002:**
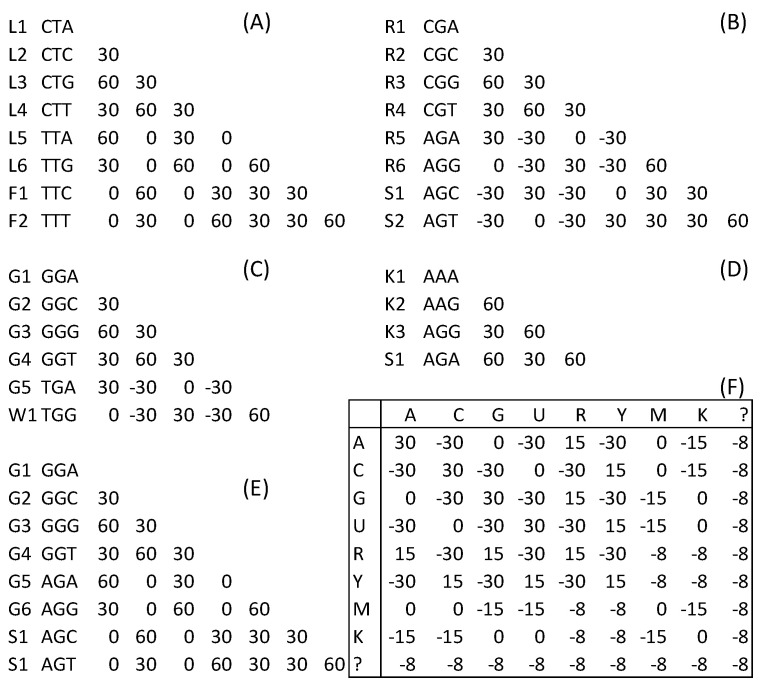
Synonymous codon families in which some synonymous codons are less similar to each other than they are to some nonsynonymous codons. (**A**) Leu codons TTA and TTG are more similar to Phe codons TTC and TTT than to synonymous Leu codons CTC and CTT. (**B**) Arg codons AGA and AGG are more similar to Ser codons AGC and AGT than to synonymous Arg codons CGC and CGT. (**C**) In genetic code 25, the Gly codon TGA is more similar to the Trp codon TGG than to other synonymous Gly codons. (**D**) In genetic codes 24 and 33, Lys codon AGG is more similar to Ser codon AGA than to the synonymous Lys codon AAA. (**E**) In genetic code 13, Gly codons AGA and AGG are more similar to Ser codons AGC and AGT than to synonymous Gly codons GGC and GGT. (**F**) The match/mismatch matrix for producing the alignment scores in (**A**–**E**). The scores involving ambiguous codes are averages, e.g., the score for A/R is the average of A/A and A/G = (30 + 0)/2 = 15.

**Figure 3 life-10-00171-f003:**
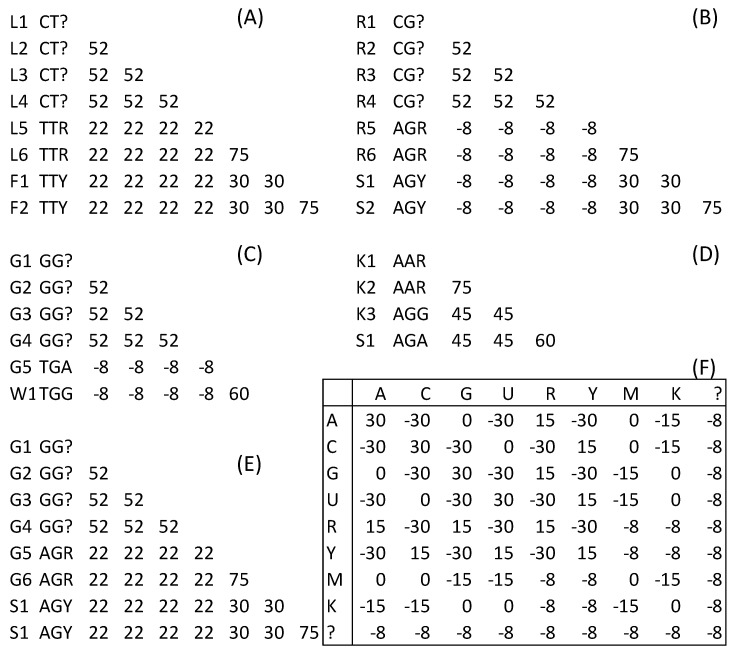
Degenerating only the third codon position. (**A**–**F**) The same as in Figure 2, but with the third codon site degenerated and the alignment scores recalculated.

**Figure 4 life-10-00171-f004:**
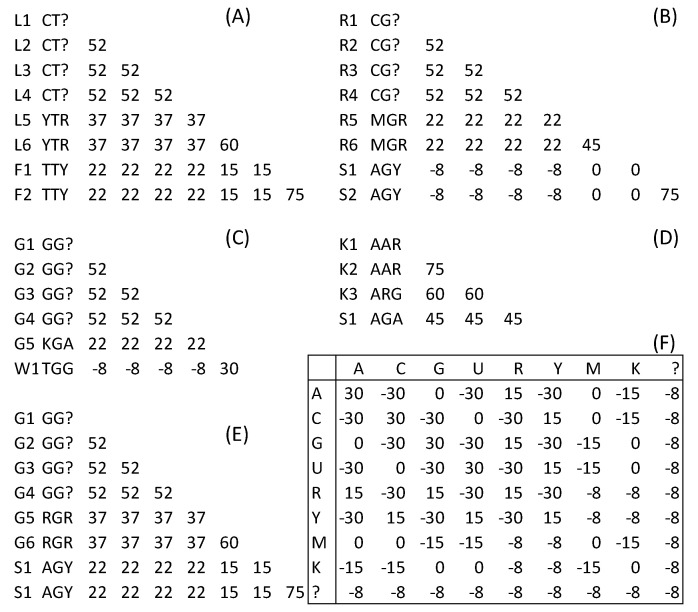
“Principled” codon degeneration. (**A**–**F**) The same as in Figure 2, but with “principled” degeneration and recalculated alignment scores.

**Figure 5 life-10-00171-f005:**
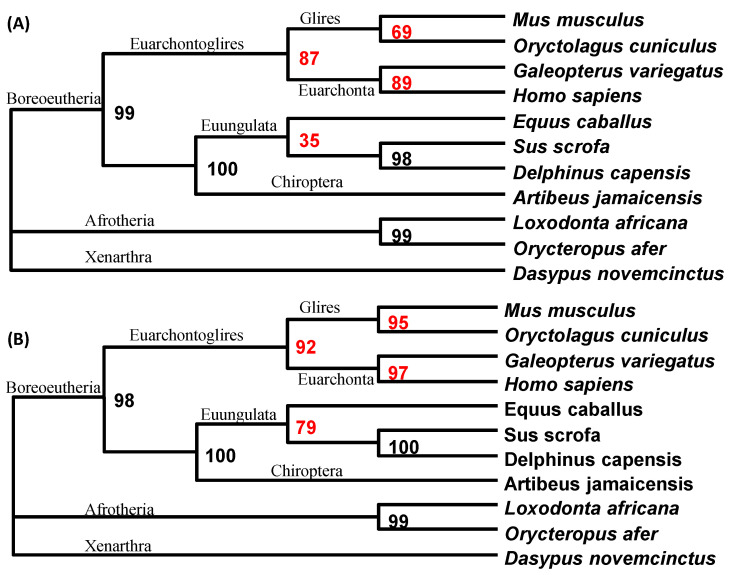
Codon degeneration improved the phylogenetic resolution of mammals. (**A**) PhyML results from sequences without codon degeneration. (**B**) PhyML results from sequences after the “principled” codon degeneration. The corresponding support values differing by ≥5% between (**A**,**B**) are highlighted in red.

**Figure 6 life-10-00171-f006:**
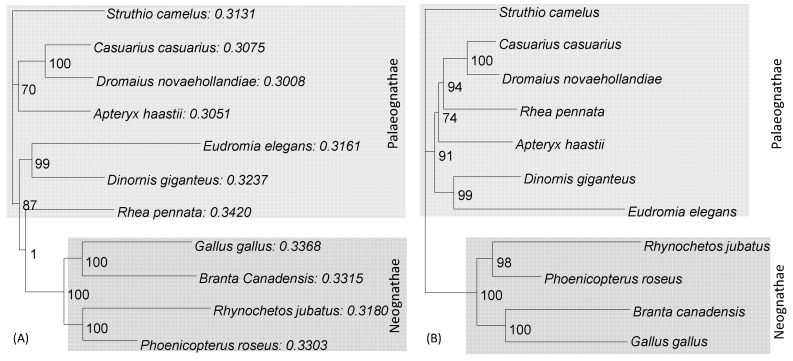
Codon degeneration removed the compositional bias. (**A**) PhyML results from sequences without the codon degeneration, leading to the wrong placement of *Rhea pennata*. The proportion of nucleotide C follows the species name. (**B**) PhyML results from sequences after the “principled” codon degeneration, which recovered the correct phylogeny.

**Figure 7 life-10-00171-f007:**
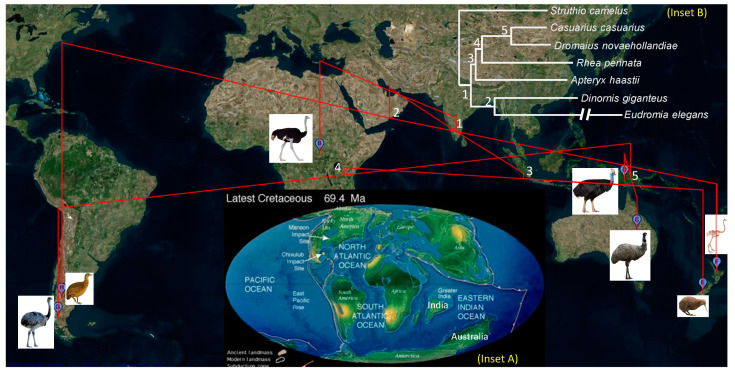
Geophylogeny of seven paleognathes, drawn using PGT software [55]. The geographic positions are approximate, with a single point representing a spatial distribution. The species images are from Wikipedia. Inset **A**: Cretaceous landmasses 69.4 million years ago (credit: US Geological Society, www.usgs.gov). Inset **B**: The phylogeny of the seven paleognathes, with node numbering identical to those on the geophylogeny.

**Figure 8 life-10-00171-f008:**
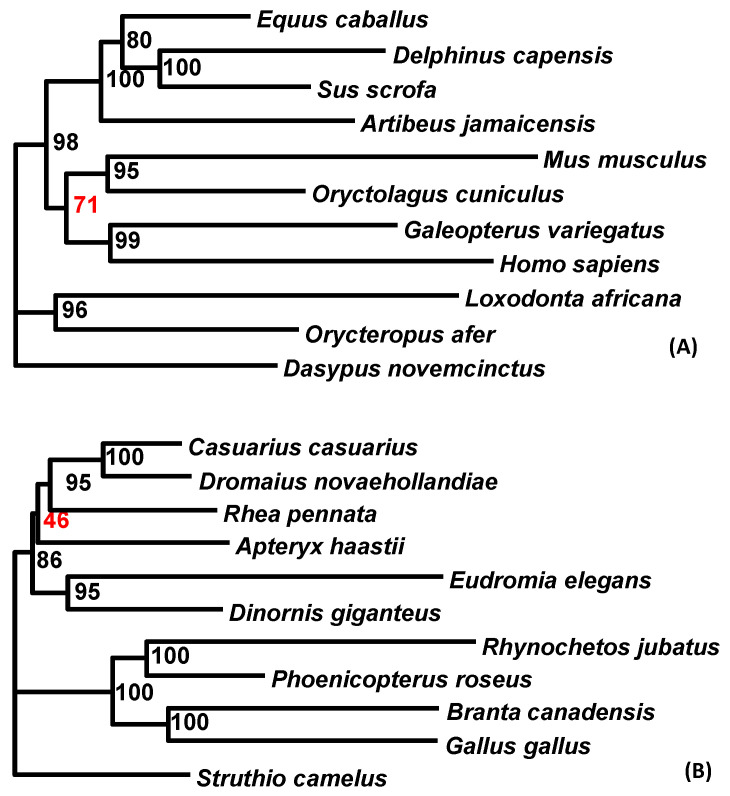
Phylogenetic reconstruction from sequences with only the third codon site degenerated. (**A**) The phylogeny from 11 mammalian species should be compared with the phylogeny in Figure 5B. (**B**) The phylogeny of 11 avian species should be compared with the phylogeny in Figure 6B.

**Figure 9 life-10-00171-f009:**
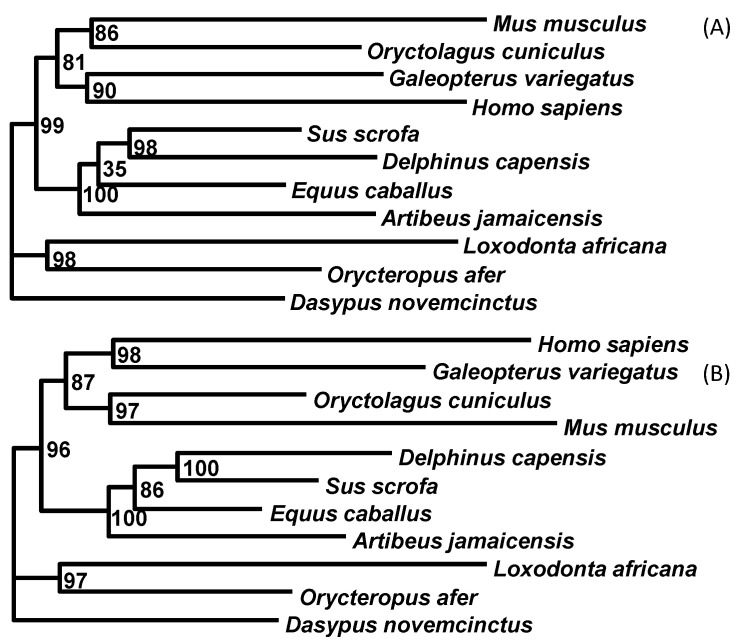
Phylogenetic reconstruction of 11 mammalian species after removing 40 codon sites featuring both UCN and AGY Ser codons: (**A**) without codon degeneration and (**B**) with “principled” codon degeneration.

**Table 1 life-10-00171-t001:** $S_{min.S}$ (minimum of the sequence similarity between synonymous codons) and $S_{max.NS}$ (maximum of the sequence similarity between nonsynonymous codons) for the illustrated codon families in Figure 2.

Case ^1^	Similarity ^2^	No ^3^	Third Only ^4^	Principled ^5^
(A)	$S_{min.S}$	0	22	37
	$S_{max.NS}$	60	30	22
(B)	$S_{min.S}$	−30	−8	22
	$S_{max.NS}$	30	30	0
(C)	$S_{min.S}$	−30	−8	22
	$S_{max.NS}$	60	60	30
(D)	$S_{min.S}$	30	45	60
	$S_{max.NS}$	60	60	45
(E)	$S_{min.S}$	0	22	37
	$S_{max.NS}$	60	30	22

^1^ (A–E) refers to the five illustrated cases in Figure 2. ^2^ Nucleotide similarity as measured using the alignment score computed with the score matrix in Figure 3F. ^3^ No codon degeneration. ^4^ Degenerated third codon positions only. ^5^ “Principled” degeneration.

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
