# Peer review of "Improving Phylogenetic Signals of Mitochondrial Genes Using a New Method of Codon Degeneration"

_life, 2020, doi:10.3390/life10090171_

Round 1

Reviewer 1 Report

The author presents a revised method for recoding protein-coding nucleotide
sequences to eliminate the effects of synonymous substitutions. The method,
which the author refers to a "principled", improves upon previous recoding
schemes by maintaining a difference between non-synonymous substitutions between Leu <-> Phe and Arg <-> Ser. The author demonstrates the effectiveness of using this method in two mitochondiral data sets.

As stated in the manuscript, the method has been previously published and
described in detail (Noah, et al. 2020), and the conclusions about how and why
the method is effective in the two demonstrating phylogenies are not novel as
they apply to previous codon-recoding methods. The current method should however
be more effective in many circumstances, by making greater use of the data. The novelty appears to be in conveying that the new method has been implemented in other genetic codes in the author's software, DAMBE.

Overall, I think the ms could be improved by discussing more widely other studies that have used similar methods, as they have drawn similar conclusions to the current study.

Specific comments:

1) To clarify my response to the bulleted question: Did you detect inappropriate
self-citations by authors? 40% (16 of 41) citations are to the authors own
work. In itself, I have no problem with this and the authors own citations might
be quite appropriate. However, in general their appear to be too few citations
to other works where these methods have been used previously and discussed.
e.g.
Criscuolo and Gribaldo 2010. BMC Evolutionary Biology 2010, 10:210
Li et al. 2014. Molecular Biology and Evolution, 31:1697–1709
Cox, et al 2014. Syst Biol 2014, 63, 272-279 is cited as a general reference to
time-heterogeneous signals, but not for using codon degenerate methods

2) Abstract: Codon degeneration decreases phylogenetic noise by aiming to
achieve two objectives: 1) alleviate bias associated with nucleotide
composition, 2) reduce differences in phylogenetic results between
nucleotide-based and amino acid-based analyses.

Codon degeneration does reduce phylogenetic noise by eliminating nucleotide
composition biases as other have noted before (Li et al, 2014; Cox et al. 2014).
But they may also improve phylogenetic accuracy by removing phylogenetic noise
(homoplasy) when the sequences are stationary - thereby reducing the likelihood of
LBA.

3) While there are various approaches to accommodate this non-stationary process [16-
21], in practice they do not increase phylogenetic resolution power [22, 23].

I strongly disagree with this statement. The author does not cite the work of
Peter Foster (Syst Biol, 53(3):485–495, 2004) and the widely used NDCH(2)
non-stationary model. Moreover, non-stationary models have been shown
to result in increased phylogenetic accuracy in many studies
(e.g. Cox et al. PNAS 105: 51, 2008; Puttick, et al, Current Biology 28:5, 2018;
Williams et al. Nature Ecology & Evolution 4:138–147, 2020)

The two citations given "[22, 23]" do not seem to appropriate to support the
argument.

Author Response

The author presents a revised method for recoding protein-coding nucleotide

sequences to eliminate the effects of synonymous substitutions. The method,

which the author refers to a "principled", improves upon previous recoding

schemes by maintaining a difference between non-synonymous substitutions between Leu <-> Phe and Arg <-> Ser. The author demonstrates the effectiveness of using this method in two mitochondiral data sets.

As stated in the manuscript, the method has been previously published and

described in detail (Noah, et al. 2020), and the conclusions about how and why

the method is effective in the two demonstrating phylogenies are not novel as

they apply to previous codon-recoding methods. The current method should however

be more effective in many circumstances, by making greater use of the data. The novelty appears to be in conveying that the new method has been implemented in other genetic codes in the author's software, DAMBE.

There are three new aspects. First, the "principled" codon generation is illustrated with various genetic codes together with the degeneration at 3rd site only and with the excessive codon degeneration in others. The "principled" codon degeneration aims to achieve two objectives: 1) minimize composition bias and 2) remove conflicting signals between nucleotide and AA sequences. I have added phylogenetic analysis with degeneration with 3rd codon only (which should remove most of the composition heterogeneity but does not remove conflicting signals between nucleotide and AA sequences).

Second, I have also added phylogenetic analysis after removing sites featuring both UCN and AGY Ser codons, before and after "principled" codon degenration.

Third, in Noah et al. (2020), we applied the method only to genes with standard genetic code. I have now implemented the method for all genetic codes, including the latest Genetic Code 33 which I have just included in DAMBE. I have also fixed a minor bug and uploaded the new version. As this special issue is on mitochondrial phylogeny, I have applied the method to analyzing mitochondrial sequences.

Overall, I think the ms could be improved by discussing more widely other studies that have used similar methods, as they have drawn similar conclusions to the current study.

I have included more relevant references.

Specific comments:

1) To clarify my response to the bulleted question: Did you detect inappropriate

self-citations by authors? 40% (16 of 41) citations are to the authors own

work. In itself, I have no problem with this and the authors own citations might

be quite appropriate. However, in general their appear to be too few citations

to other works where these methods have been used previously and discussed.

e.g.

Criscuolo and Gribaldo 2010. BMC Evolutionary Biology 2010, 10:210

Li et al. 2014. Molecular Biology and Evolution, 31:1697–1709

Cox, et al 2014. Syst Biol 2014, 63, 272-279 is cited as a general reference to

time-heterogeneous signals, but not for using codon degenerate methods

I have cited these and a number of other relevant publications.

2) Abstract: Codon degeneration decreases phylogenetic noise by aiming to

achieve two objectives: 1) alleviate bias associated with nucleotide

composition, 2) reduce differences in phylogenetic results between

nucleotide-based and amino acid-based analyses.

Codon degeneration does reduce phylogenetic noise by eliminating nucleotide

composition biases as other have noted before (Li et al, 2014; Cox et al. 2014).

But they may also improve phylogenetic accuracy by removing phylogenetic noise

(homoplasy) when the sequences are stationary - thereby reducing the likelihood of

LBA.

This is a good point. Included.

3) While there are various approaches to accommodate this non-stationary process [16-

21], in practice they do not increase phylogenetic resolution power [22, 23].

I strongly disagree with this statement. The author does not cite the work of

Peter Foster (Syst Biol, 53(3):485–495, 2004) and the widely used NDCH(2)

non-stationary model. Moreover, non-stationary models have been shown

to result in increased phylogenetic accuracy in many studies

(e.g. Cox et al. PNAS 105: 51, 2008; Puttick, et al, Current Biology 28:5, 2018;

Williams et al. Nature Ecology & Evolution 4:138–147, 2020)

I have contrasted Peter Foster's approach with a simple nucleotide RY-degeneration in Fig. 1. I believe that Cox et al was already cited in the original version. I am not comfortable to cite Puttick et al. (2018). I cited Williams et al., but only superficially. I would be more comfortable with it if I have finished analyzing the very large data sets. The phylogenetic results are more volatile and less robust than reported in the paper. I privately feel that the approach by Iwabe et al (1989, PNAS) is more informative and more tractable than that of Williams et al. However, such a discussion would distract from codon degeneration.

The two citations given "[22, 23]" do not seem to appropriate to support the

argument.

Thanks. The two references do seem out of place.

Reviewer 2 Report

This article presents a new recording scheme for mitochondrial codons. The new method is backed with a couple of seemingly successful examples in vertebrates.  It is quite well written, but it needs in my opinion some modifications and extra figures to clarify some issues, otherwise I fear that the general reader will have some problems understanding the nature and the real efficacy of the proposed recording scheme.  Indeed, based on the reading of the current manuscript, I was not able to decide if the principled recoding scheme is actually useful or not.  Accordingly, I have 3 major points/suggestions:

  1. Unclear paper structure

The structure of the paper is not clear. Manuscript starts with 1.Introduction and then moves to 2. Evolution of codon degeneration methods. I was wondering if section 2 is still part of the introduction or not. It looks partly compatible with a method section to me. Then there is section 3 which clearly belongs to results, but it is not clearly defined as results.  I advise restructuring the article having a clearer distinction among intro, methods, results, discussion.  Indeed it is better to have the datasets description in a dedicated method section, and not in the middle of the results as it is now.

  1. Clarity in explaining the principled model.

It is still not clear to me which of the codon families is actually  affected by the principled recoding scheme. I have run the recording in DAMBE and it seems to me that almost all codons are affected.  Authors should provide a new figure depicting one of the main  mitochondrial codon table (I suggest vertebrate) before and after the modification of the proposed principled codon degeneration. Also, since I think that page number is not an issue in Life, it would be better to also have the same codon table for another codon recoding strategy for example that in Regier 2010. This will strongly help the reader to understand better the recoding schemes.

  1. Efficacy of the principled degeneration method

I understood in principles the benefit of the principled method, but in practice it would be nice to see how it differs in terms of tree topology and BS to other recoding methods. Author should therefore add to both figure 3 and 4 one tree generated using the standard 3rd codon degeneration (the first method outlined in section 2) and one tree generated using the Regier2010 classical degeneration code. Furthermore, it was not clear to me if and how much the degeneration datatset provide different signal compared to the translated amino acid  dataset (after serine codons are removed). I suggest the author to further add the corresponding aa tree (I advise both before and after serine codon removal) in figure 3 and figure 4. The two datasets are rather small, therefore it should be easy to make a figure with 6 panels. With more tree comparisons the reader can judge by themself on the degree of improvement of the method.

Minor issues:

At line 7 of the introduction, the codon degeneration method is introduced. Here the author, before providing examples,  should spend a few words explaining in a very simple way what degeneration is, for example telling something like  “ it is a strategy based on the recording of some codon position in specific codons to...” In this way the author can greatly help the non specialist reader. 

Section 2. Here the author describes three methods of degenerations. The first is correctly described after “First, we may just degenerate...”. I was scrolling the text to the other methods by looking for the words “Second” and “Third” but could not find them. The second method should be introduced using the word “Second” for example as: “Second, we may degenerate Leu..” the same for the “third” method.

Figure 2: it would be helpful to have codons in this figure like it has been done in figure 3.

Figure 3 and 4: the style and position of trees is very different between the two figures. It seems they come from different articles. Author should harmonise the figures.

Page 4 second paragraph. It says: “..the two Phe codons TTA and CTT”. They are Leu codons to me,  not Phe.

Author Response

I have done perhaps the most radical revision, adding 9 figures and a table, but removed two original figures because the new ones are more informative.

This article presents a new recording scheme for mitochondrial codons. The new method is backed with a couple of seemingly successful examples in vertebrates.  It is quite well written, but it needs in my opinion some modifications and extra figures to clarify some issues, otherwise I fear that the general reader will have some problems understanding the nature and the real efficacy of the proposed recording scheme.  Indeed, based on the reading of the current manuscript, I was not able to decide if the principled recoding scheme is actually useful or not.  Accordingly, I have 3 major points/suggestions:

Unclear paper structure

The structure of the paper is not clear. Manuscript starts with 1.Introduction and then moves to 2. Evolution of codon degeneration methods. I was wondering if section 2 is still part of the introduction or not. It looks partly compatible with a method section to me. Then there is section 3 which clearly belongs to results, but it is not clearly defined as results.  I advise restructuring the article having a clearer distinction among intro, methods, results, discussion.  Indeed it is better to have the datasets description in a dedicated method section, and not in the middle of the results as it is now.

I restructured the manuscript into the conventional Introduction, Materials and Methods, Results, and Discussion. It does seem to flow better.

Clarity in explaining the principled model.

It is still not clear to me which of the codon families is actually  affected by the principled recoding scheme. I have run the recording in DAMBE and it seems to me that almost all codons are affected.  Authors should provide a new figure depicting one of the main  mitochondrial codon table (I suggest vertebrate) before and after the modification of the proposed principled codon degeneration. Also, since I think that page number is not an issue in Life, it would be better to also have the same codon table for another codon recoding strategy for example that in Regier 2010. This will strongly help the reader to understand better the recoding schemes.

I have added three figures and a table to explain the methods by using a set of genetic codes. Some readers of Noah et al. (2020) have also sent me questions about the "principled" codon degeneration. I believe that the explanation is now very clear.

Efficacy of the principled degeneration method

I understood in principles the benefit of the principled method, but in practice it would be nice to see how it differs in terms of tree topology and BS to other recoding methods. Author should therefore add to both figure 3 and 4 one tree generated using the standard 3rd codon degeneration (the first method outlined in section 2) and one tree generated using the Regier2010 classical degeneration code. Furthermore, it was not clear to me if and how much the degeneration datatset provide different signal compared to the translated amino acid  dataset (after serine codons are removed). I suggest the author to further add the corresponding aa tree (I advise both before and after serine codon removal) in figure 3 and figure 4. The two datasets are rather small, therefore it should be easy to make a figure with 6 panels. With more tree comparisons the reader can judge by themself on the degree of improvement of the method.

These contrasts suggested by the reviewer would help separating the effect of 1) composition heterogeneity (which should be largely removed by third codon degeneration) and 2) conflicting signal between nucleotide and AA sequences. I have added phylogenetic results for sequences with degeneration of 3rd codon sites only, as well as analysis with or without removing sites featuring both UCN and AGY Ser codons (Fig. 8-9). The tree generated using Regier2010's degeneration was already in Figure 1 of Regier2010, which we contrasted with the phylogenetic results after the "principled" codon degeneration (Noah et al. 2020 Evol. Bioinfo.)

By the way, the reviewer might not have noticed that this submission is for a special issue on mitochondrial DNA.

Minor issues:

At line 7 of the introduction, the codon degeneration method is introduced. Here the author, before providing examples,  should spend a few words explaining in a very simple way what degeneration is, for example telling something like  “ it is a strategy based on the recording of some codon position in specific codons to...” In this way the author can greatly help the non specialist reader.

Added. I have also included discussion on the source of composition heterogeneity: (1) shared selection among distantly related species (e.g., distantly related thermophiles), or shared tRNA pool in the host selecting for similar codon usage in viruses, (2) shared mutation (e.g., parasites in host with AT-biased spontaneous mutation), etc.

Section 2. Here the author describes three methods of degenerations. The first is correctly described after “First, we may just degenerate...”. I was scrolling the text to the other methods by looking for the words “Second” and “Third” but could not find them. The second method should be introduced using the word “Second” for example as: “Second, we may degenerate Leu..” the same for the “third” method.

Yes. Revised.

Figure 2: it would be helpful to have codons in this figure like it has been done in figure 3.

I have added Figures 2-4 to replace original Figures 1-2.

Figure 3 and 4: the style and position of trees is very different between the two figures. It seems they come from different articles. Author should harmonise the figures.

They are both from DAMBE, but one displayed support with percentage and the other with proportion. They are now "harmonized".

Page 4 second paragraph. It says: “..the two Phe codons TTA and CTT”. They are Leu codons to me,  not Phe.

Yes. Thanks.

Sincerely,

Xuhua Xia

Round 2

Reviewer 2 Report

The article improved and has a good flow. ok for pubblication.